# Arabic Wikipedia users' personalized behavior analysis considering gender gap

**Bashar Al-Shboul**[1]☯*, **Dana A. Al-Qudah**[1]☯¤, **Hadeel Boshmaf**[2]☯, **Bilal Abu-Salih**[1]‡, **Majdi Beseiso**[3]‡, **Samar Al-Saqqa**[1]‡

**1** King Abdullah II School of Information Technology, The University of Jordan, Amman, Jordan, **2** Center of Women Studies, The University of Jordan, Amman, Jordan, **3** Department of Computer Science, Al-Balqa Applied University, Al-Salt, Jordan

☯ These authors contributed equally to this work.
‡ BAS, MB and SAS also contributed equally to this work.
¤ Current address: Department of AI and Data Science, Faculty of information technology, Applied Science University, Amman, Jordan
* b.shboul@ju.edu.jo

**Data Availability Statement:** The data was downloaded from Wikipedia free dump files, an open access site for all researchers working on Wikipedia. No authorization is required. The site

## Abstract

### Introduction

As many web platforms adopt collaborative content editing models, the gender gap is addressed as one of the chief concerns in using technology to restrict content editing by one gender.

### Objective

This study aims to analyze the Arabic Wikipedia, the largest collaborative content editing platform on the Arabic web, in terms of gender behavior and differences in user activities.

### Methods

This study is the first to address the gender gap in Arabic Wikipedia, characterize users' gender through their behavior, and then address changes in characteristics over the past five years. This study analyzes parts of Arabic Wikipedia offline by linking article pages and page edit histories to user profiles of known genders.

### Results

This study reported that a gender gap exists in Arabic Wikipedia. The results reported differences over the past five years between both genders in terms of tasks and user behavior. One aspect that indicated similarity is the period of active time over months/years. Differences were observed in the reported number of increasing users, activities, responsibilities, and average actions performed.

### Conclusion

The results reveal a vast gap in terms of gender behavior in Wikipedia activities. Moreover, the results reveal that some administrative activities are disclosed to men more than to women.

was last accessed in December 2023. In addition, all extracted, then utilized contents, were compressed in an attached file containing all data required to generate tables and figures. We have uploaded our dataset/database to both: FAIRsharing and Kaggle. FAIRsharing.org: AWGS; Arabic Wikipedia Gender Study, FAIRsharing ID: https://fairsharing.org/5847, Last Edited: Saturday, September 28th 2024, 1:57, Last Editor:bashar, Last Accessed: Saturday, September 28th 2024, 1:57 https://www.kaggle.com/datasets/basharshboul1981/arabic-wikipedia-gender-study.

**Funding:** The author(s) received no specific funding for this work.

**Competing interests:** The authors have declared that no competing interests exist.

## Introduction

Owing to the availability of extensive data and the acquired role of collaborative writing, information sharing, and personalized user profiling, many users have become active and essential contributors to available e-content. Wikipedia is one of the primary platforms with the highest reputation, number of users, and content. Although the content available on Wikipedia is extensive and estimated to be more than 4.3 billion words including all English Wikipedia articles as of June 2023, Arabic content is estimated to be slightly more than 1.2 million words including all articles. Arabic user contributions remain unexplored, particularly from a gender perspective. Arabic content and users over the internet remain ambiguous because of the complexity of the language, multiple dialects, and cultural challenges of gender related to contributions of women over the internet. A few studies have been conducted on gender gap in Arabic Wikipedia. In 2013, only 11% of Arabic Wikipedia editors were identified as women. A 2016 study revealed that only 19% of Wikipedia editors in Arabic were women and only 15% of all Wikipedia articles in Arabic were about women. A 2018 study revealed that the gender gap in Arabic Wikipedia is wider in some countries than in others. This study aims to quantify gender-related issues by studying editing profiles to support Arabic Wikipedia as a more inclusive and representative resource.

The gender gap in Arabic Wikipedia is a complex issue with few indicators such as 1) the few existing women editors, 2) systemic biases in editing culture, and 3) gender preference in writing articles. As digital platforms such as Wikipedia provide powerful platforms to amplify women's voices and perspectives, they enable women to share their thoughts, experiences, and creative idea-seeking opportunities. This newfound freedom empowers women to contribute toward enriching the content landscape with diverse voices and narratives that may have been marginalized or underrepresented in conventional media [1]. This technology has the potential to free women from various gender roles and stereotypes of their roles in society by facilitating them to improve their careers in the world of work, education, and intellectuals [2, 3]. The digitization of women's roles in collaborative content writing undeniably reshapes the narrative of gender in the field and brings forth a powerful shift toward diversity and inclusivity. The digital landscape has enabled more women to work remotely as freelancers. Many women take advantage of this flexibility to engage in collaborative writing projects, contributing content from the comfort of their homes or preferred locations. This transformation not only amplifies the voices of women, but also underscores the immense talent and creativity they contribute to the world of content creation [1].

This study aims to explore the behavior of Arab users of Wikipedia, with an emphasis on gender in relation to contributions, resources, and time dedicated to their involvement. This study was conducted using information retrieval techniques and text analysis of contributors over the last five years with the aim of bridging discourses on information technology, gender, and feminism. Thus, the conceptual framework builds on creating a comprehensive and critical understanding of how gender–power dynamics influence the initiation and expansion of women's contributions to Wikipedia. Gender and feminist theories have been applied to obtain a deeper understanding of the digitization of women's roles in collaborative content writing. Cyberfeminism, an offshoot of feminism, focuses on the intersection of gender and technology, suggesting that individuals' experiences are shaped by multiple intersecting identities including gender, race, and class [4]. In the context of collaborative content writing, digitization can offer women from diverse backgrounds an equal platform to share their perspectives and experiences [1, 3].

Consequently, this study aims to answer gender-related questions by analyzing Wikipedia contributors in terms of gender. These are:

1. How does gender contribute toward characterizing user profile of Arabic Wikipedia contributors?

2. Is it possible to characterize the gender of Wikipedia contributors in terms of the size of contributions?

3. Is it possible to characterize the gender of Wikipedia contributors in terms of categories of contributions?

4. Is it possible to characterize the gender of Wikipedia contributors in terms of date and time allocated?

5. Is it possible to characterize the gender of Wikipedia contributors in terms of topics of interest?

6. What are the main page categories of interests of Arabic Wikipedia contributors from a gender perspective?

7. Is there a significant difference between genders in terms of categories of interest?

8. How do Wikipedia Arabic user contributions vary according to gender over the past five years?

This study primarily examines Arabic Wikipedia contributors by profiling users based on their gender. It highlights how contributor gender plays a significant role in characterizing their behavior, topics of interest, time allocated, and patterns of contribution. This study differs from other studies in that gender is not discussed; rather, user profiling topics are discussed in terms of gender theory with specialized input from a gender specialist to provide insights into contributors' behavior.

## Related work

With the revolution of the web, most of the content generated is user-profile-centric, that is, personalized [5] including social media, e-learning, e-government, and collaborative writing platforms. Wikipedia is an encyclopedia-scale example of collaborative writing platform, which is currently maintained by a community of volunteers using an editing system called MediaWiki. "Wikipedia is the largest and most read reference work in history and has consistently been one of the ten most popular websites, ranked 7th in October 2023" [6]

Wikipedia has been a rich source of research over the years, as many scholars have investigated the content, users, technologies, and contributions. As Wikipedia is more frequently leveraged to correct misinformation [7, 8], train machine learning tools [9], and enhance search engine results [10], the gender biases that exist on the platform can easily propagate throughout the digital landscape [11].

Studies on Wikipedia's culture of democracy are divided, as some claim that it is bureaucratic [12, 13], whereas others claim it is democratic [14–16].

As gender gap studies focusing on cyberfeminism discuss various aspects of limitations on online access and participation for women [1, 17, 18], many studies emphasize the importance of a movement to support gender equality and online coexistence among genders [19].

Despite the growing body of literature revealing gender disparities in online participation and contribution [20–22], there are limited studies on these gaps using comprehensive measures, particularly in specific contexts such as Arabic Wikipedia. This study highlights the pronounced gender gap in content contribution and page editing activities. Various explanations have been offered for these gender differences, including unequal access to resources and societal processes, such as gender socialization and Social Role Theory (SRT) [23–25]. In a

stratified society, such as the Arab region, understanding women's experiences necessitates an examination of the gender concepts, power structures, and emotional relations that govern their lives [26].

This study differs from extant literature in that it focuses on characterizing Arabic Wikipedia users through their editing behavior, indicating differences and gaps between genders, and then matching the findings with those of SRT.

## Methodology

Parts of Arabic Wikipedia were downloaded on 20/06/2023 including pages, articles, logs, categories, and modifications, among other details, with the total size exceeding 23GB. A sample Wikipedia page structure is found in Fig 1. The figure shows how page history may be included in the page metadata including: page id, time stamp, contributors, among many others. One page may have more than one contributor and may have been modified many times. Contributors' ids were used to match page contributors with their Wikipedia page edit history files.

```
<page>
<title></title>
<ns>4</ns>
<id>1295</id>
<revision>
    <id>413459</id>
    <parentid>306641</parentid>
    <timestamp>2022-11-29T10:03:04Z</timestamp>
    <contributor>
        <username> username goes here </username>
        <id>29278</id>
    </contributor>
    <model>wikitext</model>
    <format>text/x-wiki</format>
    <text bytes="4766" xml:space="preserve">
        .... page content goes here ....
    </text>
    <sha1>px4vxk1ye9fuuj2wtyhc037v9z8fwj2</sha1>
    </revision>
</page>
```

**Fig 1. Wikipedia page structure.**

Another Structured Query Language (SQL) version was downloaded to clearly understand the relationship between Wikipedia pages and other information such as categories, modification logs, and users, with the total size exceeding 21GB. The analysis was performed using a server with a Xeon Gold CPU, 96 GB of DDR4 RAM, and an NVMe SSD. Page history was the chief part of the analysis, where users, their actions, and their action dates were collected. Wikipedia history pages have the schema of (Mediawiki_history_dumps#Schema_details) page last visited (December 4, 2023). This schema table is shown in Table 1 and a short description of studied actions is listed in Table 2 in the supporting information section. According to Wikipedia, the dumps used and their content are licensed under the GNU Free Documentation License (GFDL) and the Creative Commons Attribution-Share-Alike 3.0 License [27].

The schema provides detailed information used to perform our analysis, except for gender. Therefore, a list of Arabic men and women users have been collected from the (تصنيف: رجال_ويكيبيديون ، تصنيف:نساء_ويكيبيديات translated in [28]) Wikipedia pages where usernames were matched to the users found in history pages, and the matched ones were reported in this work. After removing automatic bots, an 89% match of usernames was reported for the last five years, excluding 2023, as the log for this year was incomplete in June 2023. The remaining 11% were inactive, deleted, or banned by Wikipedia administrators for various reasons. Edited or created pages, revisions, user management, and event actions were extracted and analyzed for each user. Consequently, users have been studied from various perspectives including their possible actions as collaborators and/or editors.

## Results and discussions

The results revealed several interesting findings. The gender gap in Arabic Wikipedia indicates that there are few existing women editors as shown in the statistics. There are also systemic biases in editing culture as shown in Fig 2, and there is a gender preference in writing articles. For example, the results show that the number of users constantly editing pages on Wikipedia indicates a significant difference (t-test, $\alpha = 0.01$) between the two genders, as presented in Fig 2. Although the numbers change with similar patterns–that is, they increase and decrease similarly–the difference between the two genders remains high. At the peak of the chart (i.e., 2020), constantly editing men accounted for 76% of the total number of page edits, representing the lowest ratio between men and women. The reported ratio was approximately 80:20, with slight differences over time (except for 2020). This gender gap in digitization engagement has been reported in almost all digital transformation policies in the Arab region, including Jordan, [29] the United Arab Emirates, [30] and Qatar [31].

As digital platforms such as Wikipedia provide powerful platforms to amplify women's voices and perspectives, they enable women to share their thoughts, experiences, and creative idea-seeking opportunities. This newfound freedom empowers women to contribute toward enriching the content landscape with diverse voices and narratives that may have been marginalized or underrepresented in conventional media [1]. The statistics show the underrepresentation of women on Wikipedia as well.

This technology has the potential to free women from various gender roles and stereotypes of their roles in society by facilitating them to improve their careers in the world of work, education, and intellectuals [2, 3]. The digitization of women's roles in collaborative content writing undeniably reshapes the narrative of gender in the field and brings forth a powerful shift toward diversity and inclusivity. The digital landscape has enabled more women to work remotely as freelancers. Many women take advantage of this flexibility to engage in collaborative writing projects, contributing content from the comfort of their homes or preferred

**Table 1. Wikimedia history page schema.**

| Field class | Field name | Comment |
|---|---|---|
| Event_global | wiki_db | enwiki, dewiki, eswiktionary, etc. |
| | event_entity | revision, user or page |
| | event_type | create, move, delete, etc. Detailed explanation in the docs under #Event_types |
| | event_timestamp | When this event ocurred |
| | event_comment | Comment related to this event, sourced from log_comment, rev_comment, etc. |
| Event user | event_user_id | ID of the user that caused the event. Null if the user is anonymous or if from a revision where the user has been revision deleted. |
| | event_user_text_historical | Historical username (IP address for anonymous user) of the user that caused the event. Null for revisions where the user has been revision deleted. |
| | event_user_text | Current username of the user that caused the event. Null for anonymous users (the IP is stored in event_user_text_historical). Null for revisions where the user has been revision deleted. |
| | event_user_blocks_historical | Historical blocks of the user that caused the event |
| | event_user_blocks | Current blocks of the user that caused the event |
| | event_user_groups_historical | Historical groups of the user that caused the event |
| | event_user_groups | Current groups of the user that caused the event |
| | event_user_is_bot_by_historical | Historical bot information of the user that caused the event, can contain values name or group |
| | event_user_is_bot_by | Bot information of the user that caused the event, can contain values name or group |
| | event_user_is_created_by_self | Whether the event_user created their own account |
| | event_user_is_created_by_system | Whether the event_user account was created by mediawiki (eg. centralauth) |
| | event_user_is_created_by_peer | Whether the event_user account was created by another user |
| | event_user_is_anonymous | Whether the event_user is not registered. True for revisions where the user has been revision deleted, even if the user was actually registered. |
| | event_user_registration_timestamp | Registration timestamp of the user that caused the event (from user table) |
| | event_user_creation_timestamp | Creation timestamp of the user that caused the event (from logging table) |
| | event_user_first_edit_timestamp | Timestamp of the first edit of the user that caused the event |
| | event_user_revision_count | Number of revisions made by the event_user up to the historical time in this wiki_db (only available in revision-create events so far). For revision-create events, this includes the event itself. |
| | event_user_seconds_since_previous_revision | In revision events: seconds elapsed since the previous revision made by the current event_user_id (only available in revision-create events so far) |
| page | page_id | In revision/page events: id of the page |
| | page_title_historical | In revision/page events: historical title of the page |
| | page_title | In revision/page events: current title of the page |
| | page_namespace_historical | In revision/page events: historical namespace of the page. |
| | page_namespace_is_content_historical | In revision/page events: historical namespace of the page is categorized as content |
| | page_namespace | In revision/page events: current namespace of the page |
| | page_namespace_is_content | In revision/page events: current namespace of the page is categorized as content |
| | page_is_redirect | In revision/page events: whether the page is currently a redirect |
| | page_is_deleted | In revision/page events: Whether the page is rebuilt from a delete event |
| | page_creation_timestamp | In revision/page events: creation timestamp of the page |
| | page_first_edit_timestamp | In revision/page events: timestamp of the page's first revision. Can be before the page_creation in some restore/merge cases (see revision_is_from_before_page_creation). |
| | page_revision_count | In revision/page events: Cumulative revision count per page for the current page_id (only available in revision-create events so far) |
| | page_seconds_since_previous_revision | In revision/page events: seconds elapsed since the previous revision made on the current page_id (only available in revision-create events so far) |

*(Continued)*

**Table 1.** (Continued)

| Field class | Field name | Comment |
|---|---|---|
| user | user_id | In user events: id of the user |
| | user_text_historical | In user events: historical username or IP address of the user |
| | user_text | In user events: current username or IP address of the user |
| | user_blocks_historical | In user events: historical user blocks |
| | user_blocks | In user events: current user blocks |
| | user_groups_historical | In user events: historical user groups |
| | user_groups | In user events: current user groups |
| | user_is_bot_by_historical | In user events: Historical bot information of the user, can contain values name or group |
| | user_is_bot_by | In user events: Bot information of the user, can contain values name or group |
| | user_is_created_by_self | In user events: whether the user created their own account |
| | user_is_created_by_system | In user events: whether the user account was created by mediawiki |
| | user_is_created_by_peer | In user events: whether the user account was created by another user |
| | user_is_anonymous | In user events: whether the user is not registered |
| | user_registration_timestamp | In user events: registration timestamp of the user. |
| | user_creation_timestamp | In user events: Creation timestamp of the user (from logging table) |
| | user_first_edit_timestamp | In user events: Timestamp of the first edit of the user |
| revision | revision_id | In revision events: id of the revision |
| | revision_parent_id | In revision events: id of the parent revision |
| | revision_minor_edit | In revision events: whether it is a minor edit or not |
| | revision_deleted_parts | In revision events: Deleted parts of the revision, can contain values text, comment and user |
| | revision_deleted_parts_are_suppressed | In revision events: Whether the deleted parts are deleted to admin as well (visible only by stewards) |
| | revision_text_bytes | In revision events: number of bytes of revision |
| | revision_text_bytes_diff | In revision events: change in bytes relative to parent revision (can be negative). |
| | revision_text_sha1 | In revision events: sha1 hash of the revision |
| | revision_content_model | In revision events: content model of revision |
| | revision_content_format | In revision events: content format of revision |
| | revision_is_deleted_by_page_deletion | In revision events: whether this revision has been deleted (moved to archive table) |
| | revision_deleted_by_page_deletion_timestamp | In revision events: the timestamp when the revision was deleted |
| | revision_is_identity_reverted | In revision events: whether this revision was reverted by another future revision |
| | revision_first_identity_reverting_revision_id | In revision events: id of the revision that reverted this revision |
| | revision_seconds_to_identity_revert | In revision events: seconds elapsed between revision posting and its revert (if there was one) |
| | revision_is_identity_revert | In revision events: whether this revision reverts other revisions |
| | revision_is_from_before_page_creation | In revision events: True if the revision timestamp is before the page creation (can happen with restore events) |
| | revision_tags | In revision events: Tags associated to the revision |

locations. This transformation not only amplifies the voices of women, but also underscores the immense talent and creativity they contribute to the world of content creation [1]. The results reveal that men and women increased almost constantly in an approximately linear manner, with an average percentage of women (23%) to men (77%), as presented in Fig 3. The numbers reveal that in 2020, the number of women increased by a higher percentage than the number of men (27% and 73%, respectively). The reported results are in harmony with [32] held on Spanish language. In addition, it is shown that in the last five years, the number of contributing women on Wikipedia doubled four times from slightly over 14 thousand in year 2018 to approximately 60 thousand in year 2022, nevertheless, the increase in the number of contributing men was higher starting with slightly higher than 56 thousand in 2018 to slightly higher than 240 thousand in year 2022.

**Table 2. Wikipedia studied actions description.**

| Entity | Event type | Meaning |
|---|---|---|
| revision | create | Editing a page |
| page | create | Creating a page |
| | create-page | Page creation according to the logging table |
| | delete | Deleting a page |
| | move | Changing a page's title |
| | restore | Undeleting a page |
| | merge | Merging revisions from another page [note 2 below] |
| user | create | Registering of a new account |
| | rename | Changing the name of a user |
| | altergroups | Changing the groups (rights) of a user |
| | alterblocks | Blocking/unblocking a user |

The gender gap evident in Wikipedia contributions, particularly within the context of Arabic Wikipedia, is inadequately documented and signifies an intricate phenomenon that underscores broader societal challenges intertwined with cultural norms and gender roles. As a collaborative online encyclopedia, Wikipedia relies on voluntary contributions from a global array of contributors. Nonetheless, this study elucidates the notable underrepresentation of women among Arabic Wikipedia editors and contributors. The genesis of this gender gap is multifaceted, with cultural norms and gender roles playing pivotal roles in shaping its dynamics. Cultural norms frequently prescribe traditional gender roles and influence perceptions of appropriate behaviors for men and women. Stereotypes linking women to domestic roles or

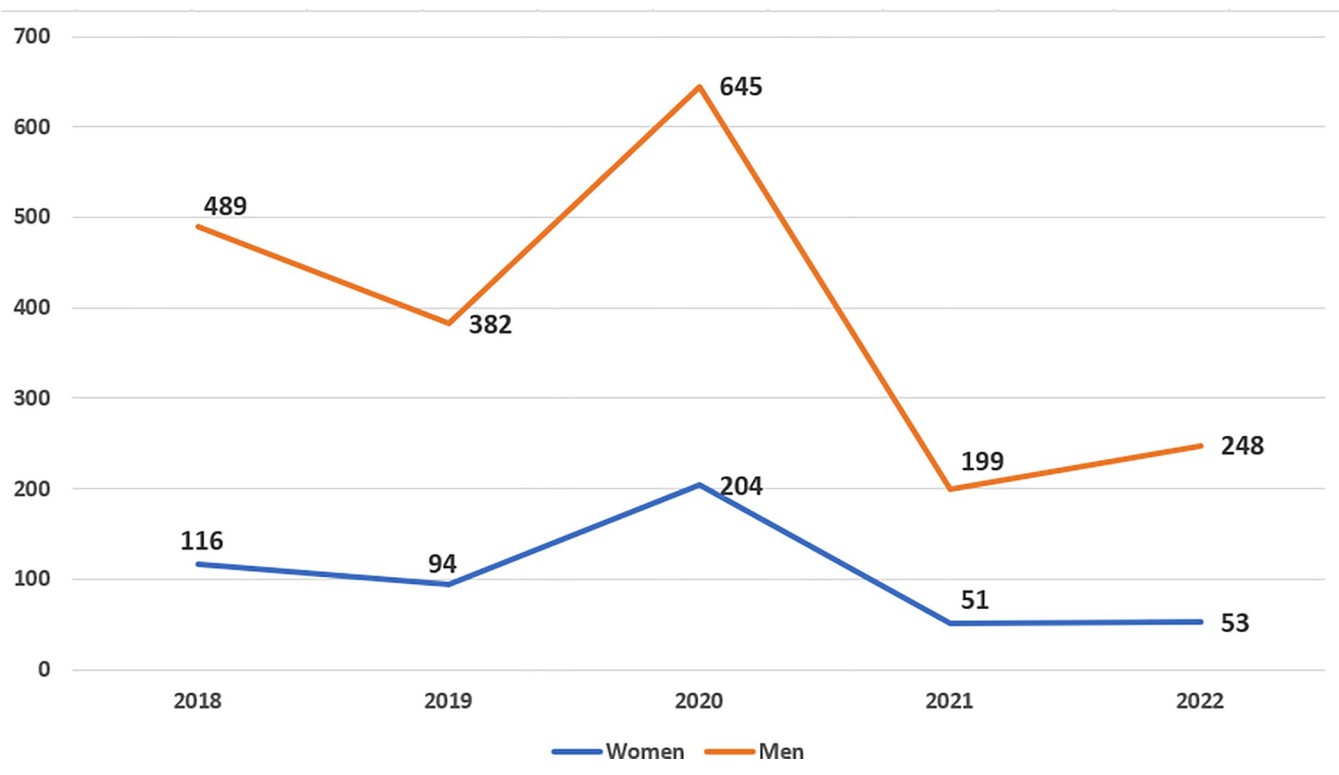

**Fig 2. Unique page edits per year by gender.**

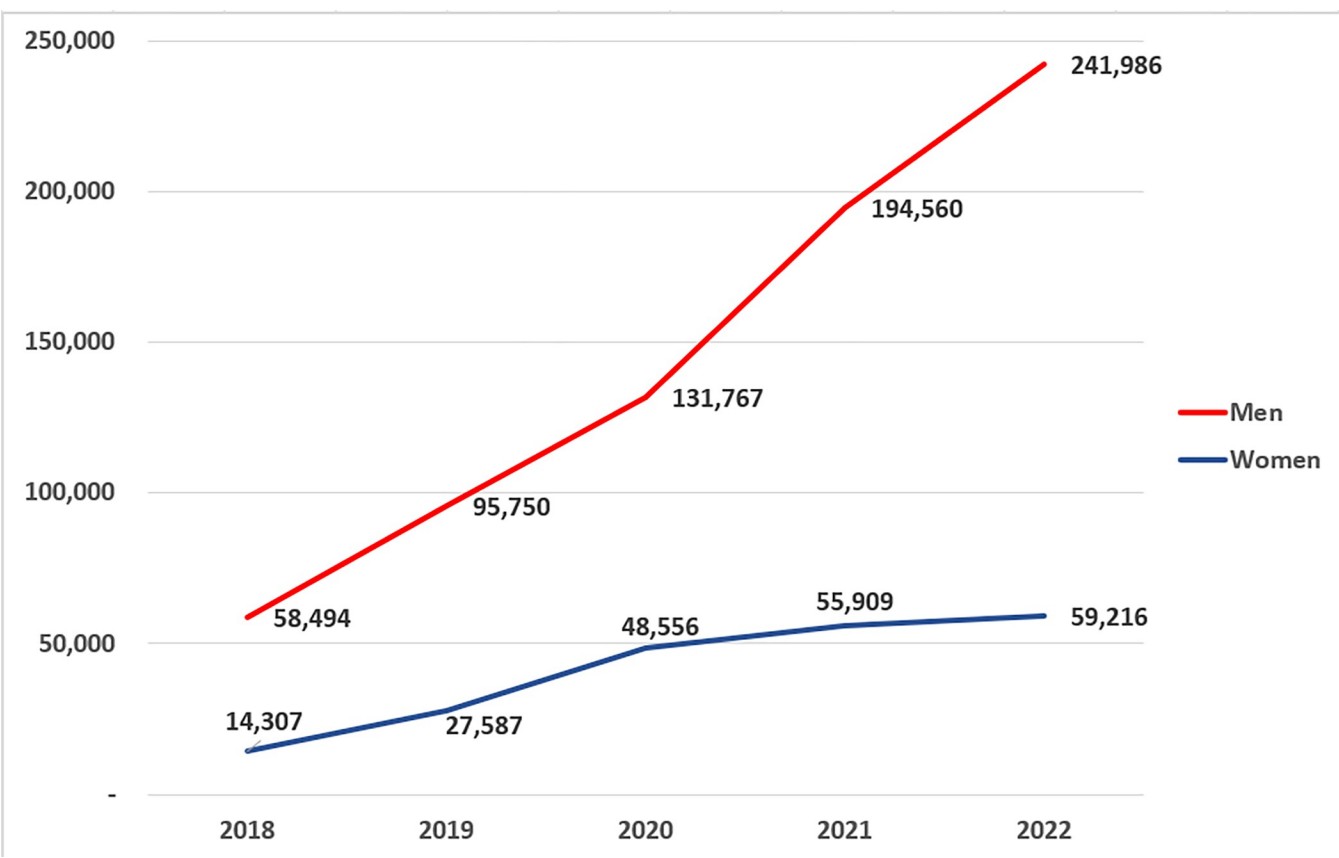

**Fig 3. The increase in Wikipedia users per year separated by gender.**

less technically oriented pursuits may deter their active involvement in fields such as technology or contributions to platforms such as Wikipedia that are perceived as predominantly male-dominated. Moreover, cultural expectations surrounding women's roles in caregiving and household management may restrict the time available for endeavors such as Wikipedia editing, with societal expectations prioritizing women's domestic responsibilities over their contributions to online platforms, thereby reinforcing traditional gender norms. Conversely, the COVID-19 pandemic provided women with distinct opportunities to engage in remote employment, work from home, and create novel job prospects that align with their role as mothers. This surge in digitalization has led to the emergence of new job opportunities in various fields. The widespread embracing of remote work practices has enabled women to more effectively reconcile their professional responsibilities with their caregiving duties, as reflected in our study results.

**Table 3. The average actions/detailed actions per year organized by gender.**

|  | Page | | | | revision | user | | Average |
|---|---|---|---|---|---|---|---|---|
|  | create-page | create | delete | Move | create | create | rename |  |
| **Male** | 42.5 | 43.5 | 2.5 | 16.0 | 436.2 | 1.1 | 8.5 | 190.2 |
| **Female** | 26.3 | 32.1 | 2.3 | 14.4 | 315.5 | 1.0 | 0.0 | 128.7 |
| **Difference** | 16.2 | 11.4 | 0.3 | 1.6 | 120.7 | 0.1 | 8.5 | 61.5 |

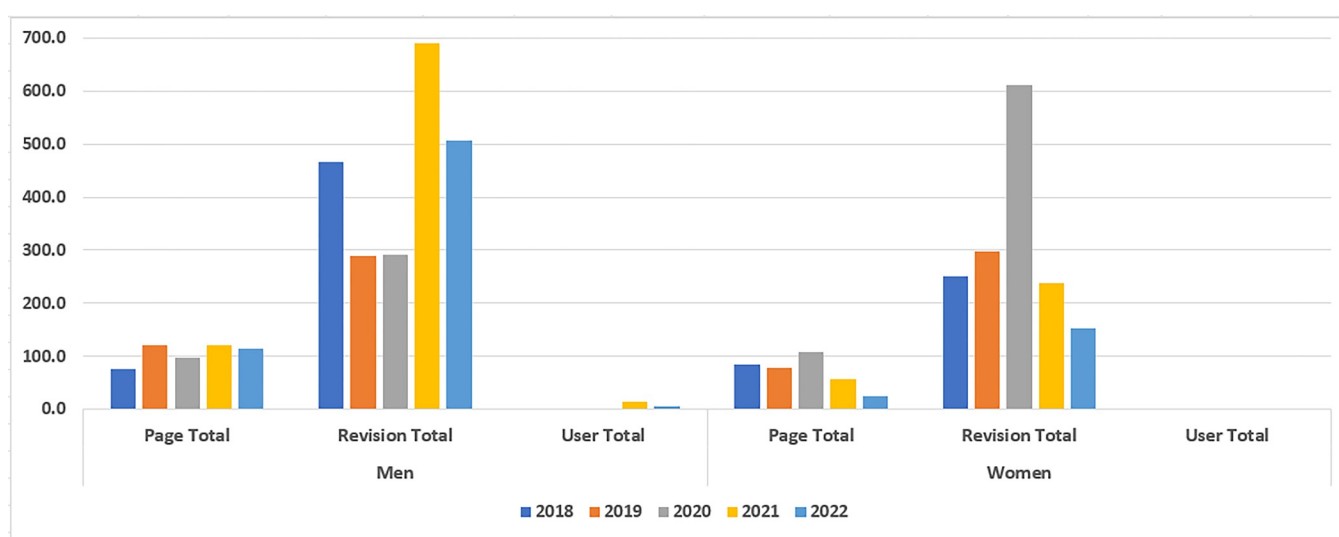

**Fig 4. A comparison between average actions (page, revision, user) per year separated by gender.**

The experiments indicated differences in both men and women's behavior toward performing actions in the last five years, as summarized in three categories: page actions (page create, delete, and move), revision actions (page create), and user actions (user create and user rename). Table 3 summarizes the average per gender in the last five years, with the differences at the bottom of the table. A t-test assuming equal means revealed that there was a significant difference at 95% (t-value = 1.823, α = 0.05), providing strong evidence that there was a significant difference between both genders. Supporting information section provides a detailed table presenting the average number of actions per year.

A summary of the **average actions per year** for men and women is presented in Fig 4. The figure displays a higher number of actions performed by men than by women, with fewer in favor of women. For example, in 2018 and 2020, women performed a higher average of page actions than men. Women also performed higher number of revisions in 2019 and 2020. Moreover, the figure demonstrates that both genders performed poorly in terms of user action over the past five years.

**Table 4. Average of action count per year (Men vs. women).**

|  | Row Labels | Page-create | Page-delete | Page-move | Revision- create | User- create | User-rename |
|---|---|---|---|---|---|---|---|
| **Men** | **2018** | 21.9 | 2.9 | 17.3 | 467.3 | 1 | 0 |
|  | **2019** | 52.4 | 1.3 | 13.5 | 289.5 | 1.1 | 0 |
|  | **2020** | 40.7 | 1.8 | 13.2 | 291.9 | 1.1 | 0 |
|  | **2021** | 53.2 | 1.8 | 14.9 | 689.7 | 2.1 | 12 |
|  | **2022** | 42.3 | 5.3 | 24.3 | 507.9 | 2 | 5 |
|  | **Average** | 42.5 | 2.5 | 16 | 436.2 | 1.3 | 8.5 |
| **Women** | **2018** | 18.52 | 6 | 25.66 | 251.44 | 1.04 | 0 |
|  | **2019** | 30.5 | 2.25 | 14.88 | 296.82 | 1 | 0 |
|  | **2020** | 44.45 | 1 | 13.44 | 611.58 | 1 | 0 |
|  | **2021** | 21.77 | 1 | 10.66 | 238.77 | 1 | 0 |
|  | **2022** | 9.27 | 0 | 4 | 153.45 | 1 | 0 |
|  | **Average** | 26.3 | 2.25 | 14.38 | 315.49 | 1.026 | 0 |

**Table 5. Sum of edits per month every year.**

|      | Jan  | Feb  | Mar  | Apr  | May  | Jun  | Jul  | Aug  | Sept | Oct  | Nov  | Dec  |
|------|------|------|------|------|------|------|------|------|------|------|------|------|
| **2018** | 4341 | 4708 | 6625 | 7795 | 5262 | 3733 | 5592 | 3494 | 4632 | 4330 | 4538 | 3444 |
| **2019** | 5559 | 2360 | 1665 | 1946 | 1872 | 3376 | 3458 | 6807 | 2696 | 2244 | 1503 | 3770 |
| **2020** | 2648 | 1388 | 1638 | 3387 | 2448 | 3513 | 3201 | 3935 | 3761 | 1824 | 2678 | 5596 |
| **2021** | 7020 | 6887 | 8452 | 5858 | 4773 | 4588 | 4184 | 3175 | 2738 | 5185 | 4663 | 5270 |
| **2022** | 3193 | 2637 | 4175 | 5407 | 5033 | 3160 | 4500 | 2164 | 3347 | 4489 | 4669 | 4652 |

## Date-based statistics

The average action count per year for each gender were reported in Table 4. In addition, it is also reported that January of each year, as shown in Table 5, is the busiest month in terms of the number of active users, followed by December, May, and April in descending order. This indicates that most users may have had more time to contribute during these months, and they may have been university students.

**Table 6. Sum of edits per day of the month.**

|      | Jan  | Feb  | Mar  | Apr  | May  | Jun  | Jul  | Aug  | Sept | Oct  | Nov  | Dec  |
|------|------|------|------|------|------|------|------|------|------|------|------|------|
| **1**  | 607  | 427  | 711  | 830  | 779  | 343  | 1040 | 737  | 264  | 255  | 663  | 697  |
| **2**  | 471  | 786  | 862  | 1043 | 482  | 411  | 769  | 817  | 1378 | 730  | 457  | 1422 |
| **3**  | 522  | 607  | 846  | 488  | 895  | 418  | 770  | 520  | 578  | 652  | 766  | 776  |
| **4**  | 806  | 545  | 892  | 1110 | 1174 | 362  | 1032 | 602  | 729  | 645  | 645  | 884  |
| **5**  | 1324 | 677  | 859  | 718  | 661  | 750  | 688  | 514  | 861  | 1014 | 759  | 584  |
| **6**  | 1097 | 612  | 556  | 1258 | 644  | 1195 | 591  | 579  | 863  | 367  | 237  | 710  |
| **7**  | 522  | 637  | 627  | 1028 | 630  | 699  | 907  | 509  | 698  | 423  | 918  | 524  |
| **8**  | 751  | 548  | 668  | 528  | 566  | 662  | 873  | 561  | 451  | 467  | 677  | 770  |
| **9**  | 811  | 1099 | 612  | 669  | 428  | 398  | 1039 | 1039 | 564  | 566  | 982  | 531  |
| **10** | 941  | 539  | 567  | 570  | 515  | 759  | 835  | 514  | 416  | 271  | 330  | 766  |
| **11** | 868  | 550  | 802  | 771  | 614  | 503  | 688  | 627  | 482  | 624  | 730  | 626  |
| **12** | 884  | 405  | 660  | 856  | 715  | 1033 | 723  | 733  | 588  | 521  | 723  | 479  |
| **13** | 696  | 409  | 560  | 1217 | 793  | 496  | 568  | 658  | 640  | 339  | 806  | 672  |
| **14** | 651  | 501  | 285  | 808  | 395  | 770  | 601  | 344  | 517  | 570  | 453  | 715  |
| **15** | 776  | 527  | 933  | 547  | 502  | 1004 | 779  | 611  | 396  | 652  | 459  | 683  |
| **16** | 748  | 657  | 1085 | 685  | 730  | 627  | 579  | 515  | 361  | 535  | 967  | 1043 |
| **17** | 634  | 651  | 645  | 397  | 522  | 480  | 646  | 740  | 361  | 463  | 634  | 995  |
| **18** | 661  | 722  | 621  | 823  | 708  | 338  | 707  | 671  | 517  | 914  | 503  | 607  |
| **19** | 707  | 859  | 1134 | 447  | 730  | 504  | 412  | 796  | 398  | 671  | 488  | 566  |
| **20** | 777  | 944  | 585  | 785  | 504  | 579  | 510  | 703  | 561  | 716  | 750  | 675  |
| **21** | 628  | 670  | 612  | 559  | 334  | 706  | 1284 | 636  | 376  | 966  | 535  | 471  |
| **22** | 440  | 592  | 654  | 668  | 304  | 611  | 485  | 541  | 556  | 338  | 699  | 684  |
| **23** | 635  | 556  | 919  | 641  | 855  | 793  | 482  | 611  | 448  | 799  | 445  | 1080 |
| **24** | 487  | 585  | 776  | 948  | 530  | 501  | 569  | 1298 | 553  | 693  | 448  | 1241 |
| **25** | 1167 | 722  | 1141 | 1055 | 577  | 506  | 410  | 916  | 1004 | 769  | 331  | 536  |
| **26** | 797  | 708  | 657  | 760  | 763  | 651  | 539  | 437  | 380  | 699  | 429  | 438  |
| **27** | 667  | 368  | 515  | 877  | 602  | 575  | 589  | 561  | 616  | 528  | 560  | 685  |
| **28** | 554  | 1002 | 658  | 861  | 1017 | 653  | 616  | 442  | 645  | 621  | 367  | 658  |
| **29** | 583  | 75   | 686  | 1542 | 667  | 587  | 448  | 305  | 371  | 425  | 757  | 1008 |
| **30** | 830  |      | 705  | 904  | 400  | 456  | 322  | 565  | 602  | 496  | 533  | 601  |
| **31** | 719  |      | 722  |      | 352  |      | 434  | 473  |      | 343  |      | 605  |

Further, as shown in Table 6, the numbers indicate that there are no significant differences (t-test, $\alpha = 0.01$ and $\alpha = 0.05$) between days of the month, as the number of active users is (488 ± 30) users, with the exception of the 31$^{st}$ of each month, as not all months have 31 days.

The number of active users has fluctuated since 2018, with a significant drop in 2022. There may be several assumptions for these changes, including marital changes and divorce rate changes during the COVID-19 pandemic; however, since biographical data for Wikipedians are not available, it is impossible to confirm these assumptions [33, 34]. The supporting materials section provides detailed tables of the day and month statistics. For example, Table 4 shows the average action count per year for each gender. For instance, the average page create action performed by men in year 2018 was 21.9 compared to 18.52 page create actions for women. In Table 5, the sum of edits per month every year is shown. This table is precisely important since it shows how women interaction changes over the year, specifically, at the times where her stereotyping roles are performed. For example, in February, June, and December the interactions decrease compared to other months due to exam periods at schools or universities and holidays.

## Conclusion

This study reported that a gender gap exists in Arabic Wikipedia. The results reported differences over the past five years between both genders in terms of tasks and user behavior. One aspect that indicated similarity is the period of active time over months/years. Differences were observed in the reported number of increasing users, activities, responsibilities, and average actions performed.

## Acknowledgments

The authors would like to thank the reviewers for their valuable insights and comments on this paper. Also, a special thanks to Editage proofreading service for their continuous support.

## Author Contributions

**Investigation:** Bilal Abu-Salih, Majdi Beseiso, Samar Al-Saqqa.

**Methodology:** Bashar Al-Shboul, Dana A. Al-Qudah, Hadeel Boshmaf.

**Writing – original draft:** Bashar Al-Shboul, Dana A. Al-Qudah, Hadeel Boshmaf, Bilal Abu-Salih, Majdi Beseiso, Samar Al-Saqqa.

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
