## [Decision Letter · Decision Letter 0]

13 Feb 2024

PONE-D-23-42255Arabic Wikipedia users’ personalized behavior analysis in the light of gender gapPLOS ONE

Dear Dr. Al-Shboul,

Thank you for submitting your manuscript to PLOS ONE. After careful consideration, we feel that it has merit but does not fully meet PLOS ONE’s publication criteria as it currently stands. Therefore, we invite you to submit a revised version of the manuscript that addresses the points raised during the review process.

**ACADEMIC EDITOR: **

Dear authors, the manuscript needs major revisions, please respond point by point to the reviewers.

Kind regards.

Dear authors, the manuscript needs major revisions, please respond point by point to the reviewers.

Kind regards.

We look forward to receiving your revised manuscript.

Kind regards,

Omar Enzo Santangelo

Academic Editor

PLOS ONE

A clean copy of the edited manuscript (uploaded as the new *manuscript* file).

3. Please change "female” or "male" to "woman” or "man" as appropriate, when used as a noun (see for instance https://apastyle.apa.org/style-grammar-guidelines/bias-free-language/gender).

4. In your Methods section, please include additional information about your dataset and ensure that you have included a statement specifying whether the collection and analysis method complied with the terms and conditions for the source of the data.

5. Please provide a complete Data Availability Statement in the submission form, ensuring you include all necessary access information or a reason for why you are unable to make your data freely accessible. If your research concerns only data provided within your submission, please write "All data are in the manuscript and/or supporting information files" as your Data Availability Statement.

Additional Editor Comments:

Dear authors, the manuscript needs major revisions, please respond point by point to the reviewers.

Kind regards.

Reviewers' comments:

Reviewer's Responses to Questions

**Comments to the Author**

1. Is the manuscript technically sound, and do the data support the conclusions?

Reviewer #1: Partly

Reviewer #2: Partly

Reviewer #3: Partly

2. Has the statistical analysis been performed appropriately and rigorously? 

Reviewer #1: Yes

Reviewer #2: No

Reviewer #3: No

3. Have the authors made all data underlying the findings in their manuscript fully available?

Reviewer #1: Yes

Reviewer #2: Yes

Reviewer #3: Yes

4. Is the manuscript presented in an intelligible fashion and written in standard English?

Reviewer #1: No

Reviewer #2: No

Reviewer #3: Yes

5. Review Comments to the Author

Reviewer #1: 1. The data used was downloaded online from the Wikipedia history page that summarizes actions. This however does not give information on the content to highlight topics about women.

You mentioned that the profiling topics will be discussed in terms of gender theory with a specialized input from a gender specialist to give a thorough analysis of the contributor's behavior. The results for this are missing.

2. The introduction section mixed up many issues that include Contributors, collaborators, Editors, content & actions. The Authors should be specific on what was studied. The results are about the editing behavior in light to being male or female is this enough to give a thorough gender perspective?

3. The Abstract should be written in the (Introduction, Methods, Results and Conclusion) format.

4. The study was not an experiment and therefore the subheading on Experimental Results should be revised.

5. The results speaking to the average actions per year between the two genders should be presented better. Using this figure is very confusing to the person interpreting the results.

6. The main text of the manuscript should be written in the (Introduction/ Background, Methods, Results, Discussion and conclusion) format.

7. The conclusion brings in a new term "assigned tasks" this has not been mentioned or discussed anywhere in the text.

8. In the abstract you indicated that "some administration activities are disclosed for males compared to women" this was not shown in the results section. Can't this be a sole reason why there is a disparity between male and female user activities shown?

9. You will need an English speaker to read through this work to improve the flow especially with sentence starters.

10. This is an interesting piece of work. Well done for the hard work.

Reviewer #2: N/A, N/A, N/A, N/A, N/A, N/A, N/A, N/A, N/A, N/A, N/A, N/A, N/A, N/A, N/A, N/A, N/A, N/A, N/A, N/A, N/A, N/A, N/A, N/A, N/A, N/A, N/A, N/A, N/A, N/A, N/A, N/A, N/A, N/A, N/A, N/A, N/A, N/A, N/A, N/A, N/A

Reviewer #3: Method section is not clear, what software was used to analysis the data. I am confused about this section. Why is there no discussion to show similar work and how the findings relate to that work? the Conclusion needs a lot of work it is not very scientific and doesn't reflect much of the results.

6. PLOS authors have the option to publish the peer review history of their article (what does this mean?). If published, this will include your full peer review and any attached files.

Reviewer #1: **Yes: **Olivia Nakisita

Reviewer #2: **Yes: **Daniel O'Keefe

Reviewer #3: No

---

## [Author Response · Author response to Decision Letter 0]

15 Apr 2024

All reviewers' comments were addressed in a response file uploaded with the manuscript.

Thank you very much for your insights and comments that helped us through this submission.

---

## [Decision Letter · Decision Letter 1]

26 Apr 2024

PONE-D-23-42255R1Arabic Wikipedia users’ personalized behavior analysis considering gender gapPLOS ONE

Dear Dr. Al-Shboul,

Thank you for submitting your manuscript to PLOS ONE. After careful consideration, we feel that it has merit but does not fully meet PLOS ONE’s publication criteria as it currently stands. Therefore, we invite you to submit a revised version of the manuscript that addresses the points raised during the review process.

We look forward to receiving your revised manuscript.

Kind regards,

Omar Enzo Santangelo

Academic Editor

PLOS ONE

Journal Requirements:

Additional Editor Comments:

Dear authors, the manuscript needs major revisions, please respond point by point to the reviewers.

Kind regards.

Reviewers' comments:

Reviewer's Responses to Questions

**Comments to the Author**

1. If the authors have adequately addressed your comments raised in a previous round of review and you feel that this manuscript is now acceptable for publication, you may indicate that here to bypass the “Comments to the Author” section, enter your conflict of interest statement in the “Confidential to Editor” section, and submit your "Accept" recommendation.

Reviewer #1: (No Response)

Reviewer #2: (No Response)

2. Is the manuscript technically sound, and do the data support the conclusions?

Reviewer #1: Yes

Reviewer #2: Yes

3. Has the statistical analysis been performed appropriately and rigorously? 

Reviewer #1: Yes

Reviewer #2: Yes

4. Have the authors made all data underlying the findings in their manuscript fully available?

Reviewer #1: Yes

Reviewer #2: Yes

5. Is the manuscript presented in an intelligible fashion and written in standard English?

Reviewer #1: Yes

Reviewer #2: Yes

6. Review Comments to the Author

Reviewer #1: 1. The abstract should be presented (Introduction, Objective, Methods, Results, Conclusion) The method used was not well explained

2. The content with the heading of "Related Work" could be used for the discussion of results which you did not include in this manuscript

3. You indicated "Experimental Results" but you did not include experiments in your methodology. Review this and be specific. downloading information is not an experiment.

4. Please include a "Discussion section" to this paper.

Reviewer #2: This version of the paper draft is a large improvement on the original. Well done. However, I feel there remains some work to be done before the paper can be published.

- The aims of the paper do not properly reflect the work performed, and are overcomplicated. Better to just specify that your aims were to (for example), "Describe the proportions of males and females among Arabic Wikipedia contributors".

- In the methods, you say "Wikipedia pages where usernames were matched to users found in history pages". The process for making this matching isn't properly explained. Is this all publicly available data? Do Wiki creators provide additional personal information about themselves (aside from gender)?

- You haven't explained any of your statistical methodology in the methods section.

- What is meant by "Page actions", "Revision actions" and "User actions" is not clear. Also, how does the same action (i.e. "page create") differ across each action?

- There's a lot of information in the results section that should be moved to the discussion. For example, the 'gender analysis' would be better placed in the discussion (though needs citations).

- Table 1 doesn't include any p-values.

- As above, the discussion is incredibly short. This is your space to interpret your results and consider your results in relation to existing global literature.

7. PLOS authors have the option to publish the peer review history of their article (what does this mean?). If published, this will include your full peer review and any attached files.

Reviewer #1: **Yes: **Olivia Nakisita

Reviewer #2: **Yes: **Daniel O'Keefe

---

## [Author Response · Author response to Decision Letter 1]

9 May 2024

Reviewer #1: 

Comment 1. The abstract should be presented (Introduction, Objective, Methods, Results, Conclusion) The method used was not well explained

Answer: An abstract with the requested structure was included for the editor to choose which is better to include in the paper. Both abstracts contain the same information.

Comment 2. The content with the heading of "Related Work" could be used for the discussion of results which you did not include in this manuscript

Answer: Some parts have been used in this modified version.

Comment 3. You indicated "Experimental Results" but you did not include experiments in your methodology. Review this and be specific. downloading information is not an experiment.

Answer: The phrase was modified to “results” only

Comment 4. Please include a "Discussion section" to this paper.

Answer: There is a section named “Results and Discussions” starting line 175.

Reviewer #2: 

This version of the paper draft is a large improvement on the original. Well done. However, I feel there remains some work to be done before the paper can be published.

Comment : The aims of the paper do not properly reflect the work performed, and are overcomplicated. Better to just specify that your aims were to (for example), "Describe the proportions of males and females among Arabic Wikipedia contributors".

Answer: The study is not about proportions only, it can characterize behavior, roles, and dedication on Wikipedia.

Comment : In the methods, you say "Wikipedia pages where usernames were matched to users found in history pages". The process for making this matching isn't properly explained. Is this all publicly available data? Do Wiki creators provide additional personal information about themselves (aside from gender)?

Answer: It was stated in the last paragraph before the conclusion section, lines 263 and 264, that : “… however, since bibliographical data for Wikipedians are not available, it is impossible to confirm these assumptions …”

The matching process is now explained in Figure 1 and page edit history schema in Table 2. In line 157: “Contributors’ ids were used to match page contributors with their Wikipedia page edit history files.”

Comment : You haven't explained any of your statistical methodology in the methods section.

Answer: All statistical methods used were basic methods, the most important part was combining this big data from Wikipedia and collect the statistics with their corresponding users and actions.

Comment : What is meant by "Page actions", "Revision actions" and "User actions" is not clear. Also, how does the same action (i.e. "page create") differ across each action?

Answer: Actions were detailed in Table 3 in the supporting materials section at the end of the paper. If the reviewers feel that the table is not enough we will gladly explain them more.

Comment : There's a lot of information in the results section that should be moved to the discussion. For example, the 'gender analysis' would be better placed in the discussion (though needs citations).

Answer: Some parts were modified and moved. This was an excellent suggestion, thank you.

Comment : Table 1 doesn't include any p-values.

Answer: p-value was mentioned in the text. Nevertheless, p=0.05 was used.

Comment: As above, the discussion is incredibly short. This is your space to interpret your results and consider your results in relation to existing global literature.

Answer: Done.

---

## [Decision Letter · Decision Letter 2]

28 May 2024

PONE-D-23-42255R2Arabic Wikipedia users’ personalized behavior analysis considering gender gapPLOS ONE

Dear Dr. Al-Shboul,

Thank you for submitting your manuscript to PLOS ONE. After careful consideration, we feel that it has merit but does not fully meet PLOS ONE’s publication criteria as it currently stands. Therefore, we invite you to submit a revised version of the manuscript that addresses the points raised during the review process.

**Dear authors, please respond point by point to the reviewers.**

**Kind regards**

We look forward to receiving your revised manuscript.

Kind regards,

Omar Enzo Santangelo

Academic Editor

PLOS ONE

Reviewers' comments:

Reviewer's Responses to Questions

**Comments to the Author**

1. If the authors have adequately addressed your comments raised in a previous round of review and you feel that this manuscript is now acceptable for publication, you may indicate that here to bypass the “Comments to the Author” section, enter your conflict of interest statement in the “Confidential to Editor” section, and submit your "Accept" recommendation.

Reviewer #1: All comments have been addressed

Reviewer #2: (No Response)

2. Is the manuscript technically sound, and do the data support the conclusions?

Reviewer #1: Yes

Reviewer #2: Yes

3. Has the statistical analysis been performed appropriately and rigorously? 

Reviewer #1: Yes

Reviewer #2: I Don't Know

4. Have the authors made all data underlying the findings in their manuscript fully available?

Reviewer #1: Yes

Reviewer #2: Yes

5. Is the manuscript presented in an intelligible fashion and written in standard English?

Reviewer #1: Yes

Reviewer #2: Yes

6. Review Comments to the Author

**Reviewer #1: **The Authors have been able to address all the comments i raised. Their work is technically sound and should therefore be considered for publication.

**Reviewer #2: **Unfortunately I've found the authors responses to my previous comments to be mostly insufficient - and in some cases, I'm more uncertain that I was previously. I return to my previous comments, and hope for more comprehensive response:

1) While you may have aimed to characterise actions performed on Wikipedia according to gender, this has only been done descriptively, using basic proportions. Even so, my previous issue was that this relatively basic aim was stated across eight separate aims, some of which are largely repeating themselves. The current eight aims could simply be summarised using a single aim, such as "We aimed to describe the interactions with Wikipedia pages across time, according to gender". Beyond something like this, I think you may be overselling the abilities of your analysis methods.

2) Based on the authors response, unfortunately, I'm now more confused by the matching process. The authors point to Figure 1, as an example of the metadata extracted - where I can't see any mention of gender specification. The authors also point to a statement in the discussion, being "...since bibliographic data for Wikipedians..." - the context of this sentence suggests you mean "biographic" rather than "bibliographic" - is this what you meant? If so, any biographic data would include gender...

Either way, the methods still don't adequate describe how the authors have extracted user data. Currently, it reads that a user name has simply been matched across pages using metadata - but how this has been used to determine gender is not explained. The wording on page 9, and the reference to citation 28 are not helpful in this regard. Can you please ensure you are simply, but explicitly, explaining the process you used here.

3) The use of "basic" statistical methods does not preclude you from providing detail of your methods in the paper. You repeatedly refer to a "significant" difference in your results section, but without prior description of your analytical methods. For example, you refer to a "significant" difference in Figure 2 - so, how did you perform this trend analysis?

4) I fee like short definitions of what is meant by your "page" and "revision" categories is warranted in the methods section. It is currently unclear, and a reader shouldn't have to go hunting in the supplementary material for this information. Further, now seeing the definitions, I wonder what is the relevance of checking gender against users creating or renaming accounts??

5) The reason I asked for p-values in Table 1 is because the text currently suggests that all means across categories were significant. Is this accurate? I repeat that it would be a good idea to state all relevant p-values.

7. PLOS authors have the option to publish the peer review history of their article (what does this mean?). If published, this will include your full peer review and any attached files.

Reviewer #1: **Yes: **OLIVIA NAKISITA

Reviewer #2: **Yes: **Daniel O'Keefe

---

## [Author Response · Author response to Decision Letter 2]

25 Jun 2024

Reviewer #1: 

Comment: The Authors have been able to address all the comments I raised. Their work is technically sound and should therefore be considered for publication.

Answer: Thank you very much for the valuable comments. It made our manuscript looks better.

Reviewer #2: 

Unfortunately, I've found the authors responses to my previous comments to be mostly insufficient - and in some cases, I'm more uncertain that I was previously. I return to my previous comments, and hope for more comprehensive response:

Comment 1) While you may have aimed to characterise actions performed on Wikipedia according to gender, this has only been done descriptively, using basic proportions. Even so, my previous issue was that this relatively basic aim was stated across eight separate aims, some of which are largely repeating themselves. The current eight aims could simply be summarised using a single aim, such as "We aimed to describe the interactions with Wikipedia pages across time, according to gender". Beyond something like this, I think you may be overselling the abilities of your analysis methods.

Answer 1) The issue was previously addressed in Line 90 as follows: “This study aims to explore the behaviour of Arab users of Wikipedia, with an emphasis on gender in relation to contributions, resources, and time dedicated to their involvement”

Afterwards, the aim was split over various research questions covering contribution size (Q2), contribution types (Q3), contribution date and time (Q4), contribution topics of interest (Q5, 6, 7), and contribution changes over the past 5 years (Q8). In addition, it was interesting to study the inverse relationship between contributions and gender (Q1).

All the questions were addressed within the text, the tables, and the figures.

Comment 2) Based on the authors response, unfortunately, I'm now more confused by the matching process. The authors point to Figure 1, as an example of the metadata extracted - where I can't see any mention of gender specification. The authors also point to a statement in the discussion, being "...since bibliographic data for Wikipedians..." - the context of this sentence suggests you mean "biographic" rather than "bibliographic" - is this what you meant? If so, any biographic data would include gender...

Either way, the methods still don't adequate describe how the authors have extracted user data. Currently, it reads that a user name has simply been matched across pages using metadata - but how this has been used to determine gender is not explained. The wording on page 9, and the reference to citation 28 are not helpful in this regard. Can you please ensure you are simply, but explicitly, explaining the process you used here.

Answer 2) True, it is biographical data as fixed in line 264 “however, since biographical data for Wikipedians…”, and to make sure it is clear, Wikipedia does not reveal personal information about their users due to privacy issues.

The matching process is described in the text as follows: “Therefore, a list of Arabic men and women users have been collected from the ( تصنيف:رجال_ويكيبيديون ، تصنيف:نساء_ويكيبيديات translated in [28]) Wikipedia pages where usernames were matched to the users found in history pages, and the matched ones were reported in this work”

To elaborate, if you go to Wikipedia, and open the categories listed in reference 28 you will find a list of Wikipedia male and female users in the two categories mentioned in the reference. So basically, the only way we could tell whether a user is male or female is by the category pages of Wikipedia listing them as males or females. Following a sample from the male category page showing that user: Ammar2010 (for example) is a male user.

Comment 3) The use of "basic" statistical methods does not preclude you from providing detail of your methods in the paper. You repeatedly refer to a "significant" difference in your results section, but without prior description of your analytical methods. For example, you refer to a "significant" difference in Figure 2 - so, how did you perform this trend analysis?

Answer 3) Fixed. With all significance mention, the method (e.g. t-test) and alpha value were mentioned.

Comment 4) I fee like short definitions of what is meant by your "page" and "revision" categories is warranted in the methods section. It is currently unclear, and a reader shouldn't have to go hunting in the supplementary material for this information. Further, now seeing the definitions, I wonder what is the relevance of checking gender against users creating or renaming accounts??

We understand that the schema of Wikipedia should be in the text; however the result will be the same since tables are separated from the text, therefore we felt that the location of the schema is not important; nevertheless, if the journal decides it is important to add the table reference inside the text, we will do it.

Answer 4) With reference to your other part of the comment, “I wonder what is the relevance of checking gender against users creating or renaming accounts??” This resembles an indicator on what privileges the users may have, since editing and naming/renaming users is a high level authority. Matching this with gender can show whether there is a difference in authority roles between the two genders. This was concluded in the abstract line 32

Comment 5) The reason I asked for p-values in Table 1 is because the text currently suggests that all means across categories were significant. Is this accurate? I repeat that it would be a good idea to state all relevant p-values.

Answer 5) Alpha value was added as answered in comment 3.

---

## [Decision Letter · Decision Letter 3]

13 Aug 2024

PONE-D-23-42255R3Arabic Wikipedia users’ personalized behavior analysis considering gender gapPLOS ONE

Dear Dr. Al-Shboul,

Thank you for submitting your manuscript to PLOS ONE. After careful consideration, we feel that it has merit but does not fully meet PLOS ONE’s publication criteria as it currently stands. Therefore, we invite you to submit a revised version of the manuscript that addresses the points raised during the review process.

**Dear Authors, the manuscript needs minor revisions.**

**Kind regards.**

We look forward to receiving your revised manuscript.

Kind regards,

Omar Enzo Santangelo

Academic Editor

PLOS ONE

Journal Requirements:

Reviewers' comments:

Reviewer's Responses to Questions

**Comments to the Author**

1. If the authors have adequately addressed your comments raised in a previous round of review and you feel that this manuscript is now acceptable for publication, you may indicate that here to bypass the “Comments to the Author” section, enter your conflict of interest statement in the “Confidential to Editor” section, and submit your "Accept" recommendation.

Reviewer #1: (No Response)

Reviewer #4: All comments have been addressed

2. Is the manuscript technically sound, and do the data support the conclusions?

Reviewer #1: Yes

Reviewer #4: Yes

3. Has the statistical analysis been performed appropriately and rigorously? 

Reviewer #1: Yes

Reviewer #4: Yes

4. Have the authors made all data underlying the findings in their manuscript fully available?

Reviewer #1: Yes

Reviewer #4: Yes

5. Is the manuscript presented in an intelligible fashion and written in standard English?

Reviewer #1: Yes

Reviewer #4: Yes

6. Review Comments to the Author

**Reviewer #1: **1. The abstract is now well structured however the conclusion is written more as the results and the results more as the conclusion. The methodology should clearly state how you got the data you used for this study.

2. The subheading "Related Work" should be added to the introduction

3. There are no results tables to support the association statistics provided

4. The results and discussion section should be presented independently and not combined

**Reviewer #4: **The manuscript is properly revised, and sufficiently improved. I think the manuscript is ready for publication.

7. PLOS authors have the option to publish the peer review history of their article (what does this mean?). If published, this will include your full peer review and any attached files.

Reviewer #1: **Yes: **Nakisita Olivia

Reviewer #4: No

---

## [Author Response · Author response to Decision Letter 3]

27 Sep 2024

Dear Reviewers,

Thank you very much for your comments. We tried to accommodate your comments as possible. We hope that this version of the paper will get your approval for publication. Please find our answers to your comments in RED.

Best Regards,

Bashar, on behalf of the authors

Reviewer #1: 

1. The abstract is now well structured however the conclusion is written more as the results and the results more as the conclusion. The methodology should clearly state how you got the data you used for this study.

DONE

2. The subheading "Related Work" should be added to the introduction

It is not possible to do that because this paper includes professors from different schools, and since it maybe acceptable to do this in one domain it is not possible to have a paper without related work section.

3. There are no results tables to support the association statistics provided

We have not provided any association data or conclusions in the paper, otherwise all statistics are provided within the supported materials.

4. The results and discussion section should be presented independently and not combined

In our paper, it is important to comment on each result table/figure directly after it was listed. This is also a common practice in the domain of IT.

Thank you very much for putting the effort to help us enhance the quality of our research. We really appreciate your reviews every time.

Reviewer #4:

 The manuscript is properly revised, and sufficiently improved. I think the manuscript is ready for publication.

Thank you very much for helping us enhance the quality of our research.

---

## [Editor Report · Decision Letter 4]

2 Oct 2024

Arabic Wikipedia users’ personalized behavior analysis considering gender gap

PONE-D-23-42255R4

Dear Dr. Al-Shboul,

We’re pleased to inform you that your manuscript has been judged scientifically suitable for publication and will be formally accepted for publication once it meets all outstanding technical requirements.

Kind regards,

Omar Enzo Santangelo

Academic Editor

PLOS ONE
---

## [Editor Report · Acceptance letter]

9 Oct 2024

PONE-D-23-42255R4 

PLOS ONE

Dear Dr. Al-Shboul, 

I'm pleased to inform you that your manuscript has been deemed suitable for publication in PLOS ONE. Congratulations! Your manuscript is now being handed over to our production team.

Kind regards, 

on behalf of

Dr. Omar Enzo Santangelo 

Academic Editor

PLOS ONE